# *Wolbachia* reduces virus infection in a natural population of *Drosophila*

Rodrigo Cogni [1✉], Shuai Dominique Ding[2,3], André C. Pimentel[1,3], Jonathan P. Day[2] & Francis M. Jiggins [2✉]

*Wolbachia* is a maternally transmitted bacterial symbiont that is estimated to infect approximately half of arthropod species. In the laboratory it can increase the resistance of insects to viral infection, but its effect on viruses in nature is unknown. Here we report that in a natural population of *Drosophila melanogaster*, individuals that are infected with *Wolbachia* are less likely to be infected by viruses. By characterising the virome by metagenomic sequencing and then testing individual flies for infection, we found the protective effect of *Wolbachia* was virus-specific, with the prevalence of infection being up to 15% greater in *Wolbachia*-free flies. The antiviral effects of *Wolbachia* may contribute to its extraordinary ecological success, and in nature the symbiont may be an important component of the antiviral defences of insects.

[1] Department of Ecology, University of São Paulo, São Paulo, Brazil. [2] Department of Genetics, University of Cambridge, Cambridge, United Kingdom. [3]These authors contributed equally: Shuai Dominique Ding, André C. Pimentel. ✉email: rcogni@usp.br; fmj1001@cam.ac.uk

Wolbachia is an alphaproteobacterium that lives within the cytoplasm of arthropod cells and is maternally transmitted. It infects approximately half of arthropod species[1], and many strains manipulate host reproduction, most commonly by inducing cytoplasmic incompatibility (CI)[2]. CI allows Wolbachia to invade populations by causing embryonic mortality when uninfected females mate with infected males, hence conferring a selective advantage to infected females[3,4]. Wolbachia can also protect Drosophila species against RNA viruses[5,6]. Combined with Wolbachia's ability to invade populations due to CI, this provides a way to modify natural insect populations to make them resistant to viral infections. Wolbachia has been transferred from Drosophila to the mosquito Aedes aegypti, where it limits the replication of the dengue virus as well as chikungunya, yellow fever, Zika and West Nile viruses[7–10]. When Wolbachia-infected mosquitoes were released into the wild, the bacterium spread through the mosquito populations by CI[11], and large field trials have shown substantial reductions in dengue incidence in the human population[12,13].

While the antiviral effects of Wolbachia have great value in the field of public health, their ecological importance is far from clear. As Wolbachia is estimated to infect 52% of arthropod species[1], it may be a major component of antiviral defences in nature. However, studies on the antiviral effects of Wolbachia have largely been performed under laboratory conditions and frequently with artificial routes of infection. Wolbachia protects wild mosquitoes against the dengue virus, but here Wolbachia has been artificially transferred into the mosquito, resulting in an activation of immune defences that is not typical of natural host-Wolbachia associations[14]. Furthermore, two studies of natural populations of Drosophila melanogaster have failed to find evidence of Wolbachia protecting infected insects against viral infection[15,16], so there is currently no evidence that Wolbachia is a natural antiviral defence of insects[14,17].

If antiviral protection is present in nature, Wolbachia may frequently be a mutualist that defends its host against infection. This may explain why Wolbachia strains that do not cause CI and have no obvious phenotypic effect can invade and be maintained in populations. For example, the Wolbachia strain wAu spread through Australian populations of Drosophila simulans despite not causing CI[18]. In the same host species, the wRi Wolbachia strain has evolved to become a mutualist, but the cause of the fitness benefit is unknown[19]. The benefits provided by antiviral protection could also allow CI-inducing strains of Wolbachia to invade new populations and species. Theory predicts that CI can only invade when local infection frequencies become sufficiently high to offset imperfect maternal transmission and infection costs[20]. However, recent data suggested that Wolbachia can spread from arbitrarily low frequencies[18]. This can be explained if there is a fitness advantage for the host caused by Wolbachia, which may be its antiviral effects.

## Results

### Wild Drosophila melanogaster harbour a diverse community of viruses.
We collected 1014 male D. melanogaster from an orchard in Connecticut, USA and extracted RNA from single flies. To characterize the diversity of viruses in this population in an unbiased way, we pooled RNA from groups of 23 flies to generate 40 RNAseq libraries. These were mapped to the published genome sequences of D. melanogaster, the Wolbachia strain wMel and known Drosophila viruses. The unmapped reads were then assembled to identify novel Drosophila-associated viruses (see methods for inclusion criteria).

We identified 30 viruses associated with D. melanogaster in this population (Fig. 1). There was a wide range of abundance, with ~260,000 times more reads from the most abundant virus relative to the least abundant virus (Fig. 1). Seventeen of the viruses we identified, including the twelve most abundant ones, have previously been described as infecting Drosophila melanogaster[16,21–27].

We identified thirteen viruses that have not been associated with D. melanogaster before. We reconstructed the phylogeny of these viruses based on predicted protein sequences, and refer to them by the name of the virus family (Supplementary Fig. 1). One of these viruses belongs to the Flaviviridae and is closely related to Hermitage virus from Drosophila immigrans[22]. One virus from the order Picornavirales is closely related to Basavirus sp. A novel virus belonging to the Tymoviridae is closest to Bee Macula-Like virus 2, which has been detected in several wild bee species[28]. Four novel viruses identified within the Totiviridae clustered with Ahus virus from Culex mosquitoes[29], Keenan toti-like virus from the fly Sarcophaga impatiens[30] and Leishmania RNA virus from a trypanosome. One virus was a negative-sense RNA virus related to Drosophila unispina virus 1[27]. Five viruses belong to the Narnaviridae, and these were related to a virus from a fungus (Plasmopara viticola lesion associated narnavirus 2), an arthropod (Serbia narna-like virus 4-like) and a trypanosome (Leptomonas seymouri RNA virus-like). As viruses will be present in the food, environment and pathogens of flies, we would caution that the presence of these viruses in our samples does not mean they infected D. melanogaster, although the close relationship of many of them to other arthropod viruses suggests that some do (Supplementary Fig. 1).

We used our RNAseq data to design PCR primers that matched the eleven viruses present in all our libraries, and tested the panel of 1014 individual flies for infection by quantitative PCR. Viral infection is common, with 93% of flies infected with at least one virus ($N = 938$, including data only from samples tested for all 11 viruses). This infection rate was driven by the high prevalence of Galbut virus and Vera virus, which infect 68% and 75% of flies respectively (Fig. 1). These belong to the Partitiviridae, a family of viruses with segmented double-stranded RNA genomes. Galbut virus, which has previously been reported to infect most wild D. melanogaster[16], is efficiently vertically transmitted through both males and females, likely explaining its high prevalence[21]. Seven other viruses infected over 10% of flies (Fig. 1). The viruses that we assayed by PCR cover a diversity of taxonomic groups, including a double-stranded DNA virus (Kallithea virus), a negative-sense RNA virus (Drosophila melanogaster sigmavirus), two dsRNA viruses (Vera and Galbut viruses) and six positive-sense RNA viruses (La Jolla, Craigies Hill, Motts Mill, Nora, Dansoman, Thika, Kilifi and Drosophila A viruses).

### Wolbachia protects wild flies against viral infection.
Seventy-one percent of the flies carried Wolbachia ($N = 1014$), and these flies were infected with fewer viruses. Wolbachia-free flies were infected with a mean of 2.85 viruses, which is 15% more than the number of viruses detected in Wolbachia-infected flies (2.48 viruses; Wilcoxon rank sum test: $W = 10,1030$, $p = 0.0005$), suggesting the Wolbachia is protecting flies against infection in nature.

We estimated the prevalence of each virus in Wolbachia-free and Wolbachia-infected flies, and found there are no cases where the symbiont completely blocks viral infection (Fig. 2A). To quantify the level of protection we calculated the risk that a Wolbachia-free fly was infected with a virus relative to the risk of a fly carrying Wolbachia (Fig. 2B). In 9 out of 10 cases the risk of infection was greatest in Wolbachia-free flies (Fig. 2A, B), and for two viruses this effect was statistically supported (Fig. 2A, B;

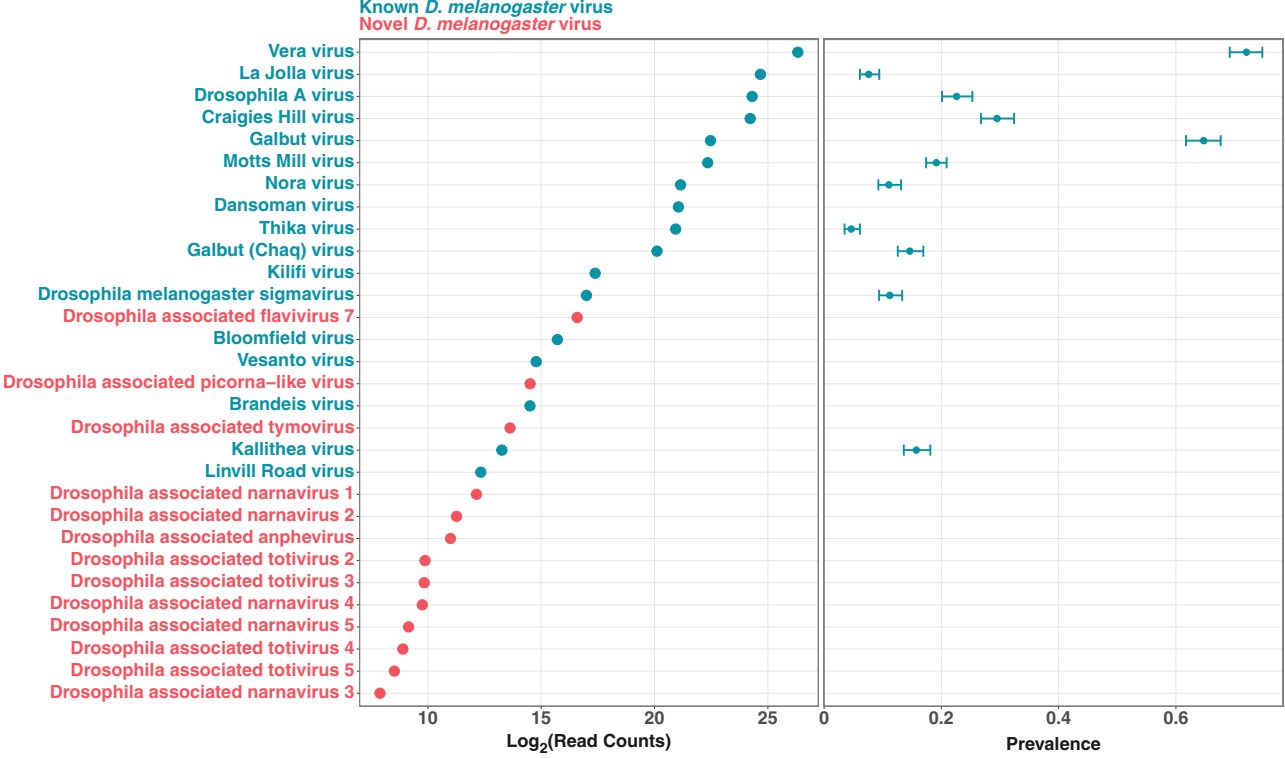

**Fig. 1 Viruses associated with wild _D. melanogaster_.** The total number of RNAseq reads that map to each virus (left). The prevalence of selected viruses estimated using quantitative PCR to test single flies for infection (right). Error bars are 95% confidence intervals.

$p_{\text{mcmc}} < 0.001$). These were a positive-sense RNA virus—Motts Mill virus—where the _Wolbachia_-free flies were 2.73 times more likely to be infected, and the dsRNA partitivirus Vera virus, where _Wolbachia_-free flies were 1.19 times more likely to be infected (Fig. 2B). For both of these viruses, we repeated the PCR tests of all the samples using an independent set of primers to verify these results (Supplementary Fig. 2a and b).

As well as reducing the likelihood that flies are infected, _Wolbachia_ could reduce viral loads in infected flies. To investigate this, we examined viral loads among the virus-infected flies. For nine of the ten viruses, there is no significant difference between the _Wolbachia_-infected and _Wolbachia_-free flies (Supplementary Fig. 3; $p > 0.01$). However, Galbut virus loads were significantly lower in the presence of _Wolbachia_ (Fig. 2C; $p = 0.0007$). Comparing the distribution of viral loads, it is clear that this is caused by a minority of flies with strongly reduced viral loads in the _Wolbachia_-infected flies, while most individuals have similar loads (Fig. 2C). Furthermore, this result still holds if the viral load was not normalised to _rpl32_ mRNA levels, indicating that it is not an artefact of _Wolbachia_ affecting the expression of the reference gene we used ($F = 14.47$, d.f. $= 1632$, $p = 0.0002$).

## Discussion

We have found that _Wolbachia_ protects wild _Drosophila_ against viral infection, with _Wolbachia_-infected flies carrying on average 0.37 fewer viruses. As viruses are common in natural insect populations, this phenotype may benefit many _Wolbachia_-infected insects and partly explain the extraordinary ecological success of _Wolbachia_. If the magnitude of this benefit is sufficient to outweigh the fitness cost of carrying _Wolbachia_, the symbiont will become a mutualist that can invade populations in the absence of other phenotypes. Establishing whether this is the case is particularly important as the _Wolbachia_ strains that provide the greatest anti-viral protection tend to be associated with the

highest fitness costs, as both traits depend on the density of _Wolbachia_ in insect cells[31]. However, even if the benefits of antiviral protection are insufficient to make _Wolbachia_ a mutualist and there remains a net fitness cost, then the antiviral phenotype can still reduce this cost, making it more likely that _Wolbachia_ can invade populations as a reproductive parasite[20].

The effect of _Wolbachia_ on host fitness will depend not only on the reduction in viral prevalence and titre, but also on how harmful virus infection is to the fly. Of the three viruses affected by _Wolbachia_, only the phenotypic effects of Galbut virus infection have been reported. Under laboratory conditions this virus had only very modest effects on lifespan and fecundity[32]. If we speculate that these results hold for other viruses affected, and given that _Wolbachia_-infected flies carrying ~0.37 fewer viruses, the magnitude of any fitness benefit might be so small as to have minimal impact on _Wolbachia_ dynamics. However, harsh competitive conditions can increase the cost of infection, and these may be common in the field. For example, flies infected with the _Drosophila melanogaster sigmavirus_ appear healthy in the laboratory. However, in the field or under competitive laboratory conditions it is estimated to reduce fitness by 20–30%[33,34]. If this was the case for the viruses affected by _Wolbachia_, then the benefits of antiviral protection could be as high as 10%. This is comparable to the fitness benefit of _w_Au that allowed it to invade populations of _Drosophila simulans_ in the absence of CI[18].

An important caveat to this study is that we only investigated males, as we could not reliably morphologically identify female _D. melanogaster_ to the species level. However, because _Wolbachia_ is maternally transmitted, it is antiviral protection in females that will have the greatest effect on the symbiont's fitness and population dynamics. Therefore, an important question for the future is whether similar levels of antiviral protection are seen in female hosts.

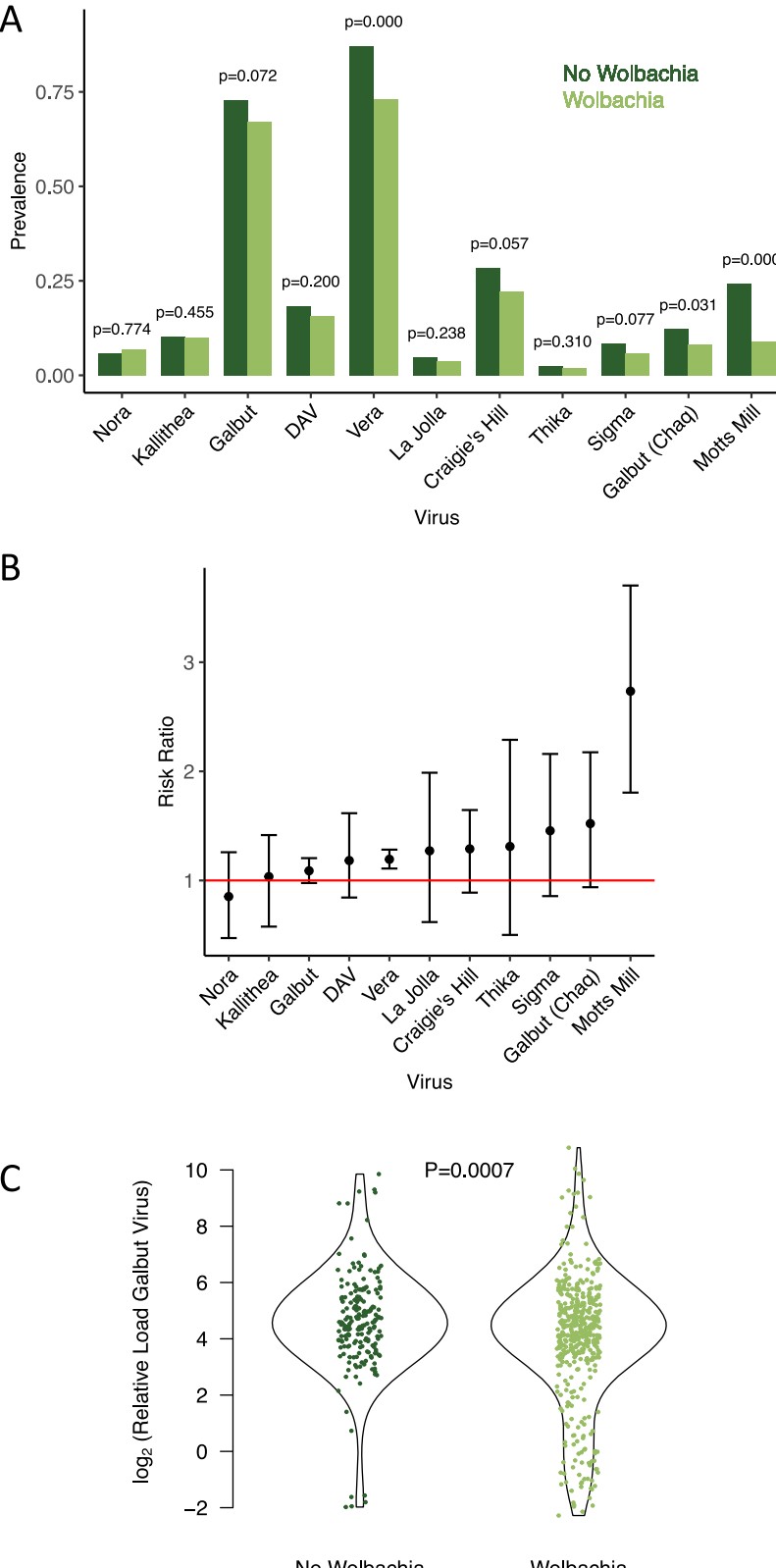

**Fig. 2 Viral prevalence and load in *Wolbachia*-free and *Wolbachia*-infected flies. A** The prevalence of viruses in male *D. melanogaster*. The bars are the posterior means of the random effect estimates of a glm. The *p* values are posterior probabilities that the prevalence differs in *Wolbachia*-free and *Wolbachia*-infected flies, estimated from the glm. **B** The risk of viral infection in *Wolbachia*-free flies relative to *Wolbachia*-infected flies. Values above 1 indicate that *Wolbachia*-free flies are more likely to be infected. The points are posterior means and the error bars are 95% credible intervals estimated from a glm. **C** Viral load of Galbut virus in flies with and without *Wolbachia*. Viral load is measured by quantitative PCR relative to the *Rpl32* mRNA. The *P*-value is the result of a one-way ANOVA.

Our results contrast with three previous that failed to find any effect of *Wolbachia* on the natural viral community of *Drosophila*. The first of these was a study designed to characterize the diversity of viruses infecting *D. melanogaster* and *D. simulans*, and the authors suggest their sampling design means they have low power to detect the effects of *Wolbachia*[16]. The second study investigated *D. melanogaster*, but used considerably smaller sample sizes than us and reared the flies for one or more generations in the laboratory at 19 °C before testing them[15]. It was later discovered that the antiviral effect of *wMel* is greatly reduced at this temperature[35]. Finally, another study investigated *D. simulans* but used comparatively small sample sizes that are unlikely to detect effects of the size we observed[36].

The microbiome plays a key role in protecting animals against infection, and in insects, this role is frequently played by specialized heritable endosymbionts that function alongside the immune system as an integral component of the animal's defences against infection[37]. For the first time, our results demonstrate *Wolbachia* naturally protects wild insects against infection and should therefore be regarded as a defensive symbiont. Because *Wolbachia* is so common in terrestrial arthropods[1] it may be an important component of antiviral defence in many species. This has the potential to affect the population biology of beneficial and pest insects, disease transmission by vector species, and the evolution of insect immune defences[38].

## Methods

**Field collection**. Flies were collected at Lyman Orchards in Middlefield, CT, USA, a common field site to collect natural *Drosophila melanogaster* populations[39,40]. From 4 to 6 September 2018, we collected a total of 1014 *D. melanogaster* males by aspirating and netting over fermenting dropped peaches. We collected males as they can be identified to species level morphologically and individually preserved them in RNAlater™ reagent a few hours after field collecting.

**RNA preparation and *Wolbachia* screening**. RNA was isolated from single flies using TRIzol™ (ThermoFisher, 15596018) extraction as previously described[41]. RNA pellets were re-suspended in 10 µl nuclease-free water (ThermoFisher, AM9930) and stored at −80 °C. Half of the RNA from each fly was saved for library preparation and half was reverse transcribed with Promega GoScript reverse transcriptase and random hexamer primers. cDNA was diluted 1:10 with nuclease-free water. RT-qPCR was performed on an Applied Biosystems StepOnePlus system using Sensifast Hi-Rox Sybr kit (Bioline) with the following PCR cycle: 95 °C for 2 min followed by 40 cycles of: 95 °C for 5 s followed by 60 °C for 30 s. Each sample was tested for *Wolbachia* infection by amplification of a segment of the gene *atpD* by RT-qPCR using primers CCTTATCTTAAAGGAGGAAA and AATCCTTTATGAGCTTTTGC[31]. To normalise estimates of *Wolbachia* and virus loads we also amplified the fly gene *RpL32* using primers TGCTAAGCTGTCGC ACAAATGG and TGCGCTTGTTCGATCCGTAAC[42].

**Library preparation and RNA sequencing**. Single fly RNA samples were combined into 40 different pools, each pool contained samples from 23 individual flies to give a total volume of 69 µl per pool. The RNA from each pool was quantified using Qubit RNA HS assay kit (ThermoFisher, Q32852). RNAseq libraries were prepared from each RNA pool as follows: Ribosomal RNA was depleted using a Ribo-Zero Gold rRNA Removal Kit (Human/Mouse/Rat) (Illumina, MRZG12324). Between 620 ng of RNA for the lowest and 1800ng of RNA for the highest sample in a total volume of 28 µl was used. To this was added 8 µl of Ribo-zero removal solution and 4 µl of Ribo-Zero reaction buffer. The protocol was followed according to the manufacturer's recommendation. The rRNA-depleted RNA was cleaned up using ethanol precipitation and the resulting pellet was re-suspended in 5 µl of nuclease-free water.

RNAseq libraries were prepared using the NEB Next Ultra II Directional RNA Library Prep kit for Illumina (New England Biolabs, E7760L) according to the manufacturer's recommendations. All 5 µl of the rRNA-depleted sample was used for each library. Adapters used were from KAPA Single-Indexed Adapter kits KK8701 and KK8702, the 30 µM stock was diluted to 1.87 µM before use. Eight cycles of PCR were used to amplify the libraries. Libraries were quantified using Qubit HS DNA quantification kit (ThermoFisher, Q32854). The final concentration of libraries was 11–29 ng/µl in a volume of 20 µl. The quality of the libraries was assessed using Bioanalyzer HS DNA kit (Agilent, 5067–4626) according to the manufacturer's instructions. The relative quantity of the libraries was ascertained using qPCR: 3 × 1:1000 dilutions were made from each library by adding 1 µl of library to 1 ml of 10 mM Tris-HCl pH8.0 with 0.05% Tween 20. Two microlitres of each dilution was used in a qPCR reaction using primers,

IS5.reamp.P5: AATGATACGGCGACCACCGA and IS6.reamp.P7: CAAGCAGA AGACGGCATACGA[43]. Libraries were normalized to the concentration of the lowest in the pool by diluting in 0.1x TE buffer and combined into three separate pools of 13 or 14 libraries. The multiplexed library pools were quantified by Qubit HS DNA as above and assessed for quality and average fragment length using a Bioanalyzer HS DNA kit as above. The concentration of each pool was calculated and then diluted to 20 nM by adding the appropriate quantity of 0.1x TE buffer before sequencing. Paired-end RNA sequencing reads from 40 libraries were obtained. Libraries were sequenced on three lanes of the Illumina HiSeq4000 with paired end 150 bp reads. Quality control of the raw RNA sequencing reads was implemented with TrimGalore-0.6.0 (http://www.bioinformatics.babraham.ac.uk/projects/trim_galore/).

**Mapping to published genomes**. The bioinformatic analyses are summarized in Supplementary Fig. 4. Trimmed reads were mapped to combined genomes of *Drosophila melanogaster*, *Wolbachia* strain *w*Mel and viruses isolated from or associated with flies in the genus *Drosophila* (Supplementary Fig. 4; Round 1 Mapping). To account for genetic variation in the viral population, the viral sequences included all the sequences deposited in GenBank. Mapping was carried out with STAR-2.6.0 with default settings[44]. Uniquely and multiple mapped reads were collected and counted for *D. melanogaster*, *Wolbachia* and each virus. Multiple mapped reads were counted only once as a randomly selected location where they had mapped.

**Virus discovery**. To reduce the size of the dataset, unmapped reads from all libraries were pooled and mapped to Ribosomal RNAs (rRNA) database downloaded from SILVA[45], including both SSU and LSU datasets, using bowtie2-v2.3.5.1[46]. The rRNA reads were removed from the pooled reads. Trinity-v2.8.4[47] was then used to assemble transcript sequences from the pooled RNAseq reads with minimum contig length set at 200 nucleotides. Assembled contigs with open reading frames no shorter than 30 amino acids identified by TransDecoder (https://github.com/TransDecoder/TransDecoder) were collected and subsequently blasted to NCBI non-redundant protein database and viral non-redundant protein database using DIAMOND blastx[48]. Contigs with blast top results corresponding to viral origins in both databases were identified as candidate viral contigs, and were selected to be assembled into longer contigs using Sequencher 4.5 (http://www.genecodes.com), followed by manual curation (Sequencher contigs).

These candidate viral contigs were once more queried against the NCBI non-redundant protein database using DIAMOND blastx to identify closely related viruses for inclusion in phylogenetic analyses. Novel viruses where the top blast hit in GenBank did not infect eukaryotes were excluded from downstream analyses. Where available, RNA-dependent RNA-polymerase protein sequences of these related viruses were then used to construct phylogenetic trees. Multiple sequence alignment was done using the M-Coffee mode in T-Coffee[49]. Phylogenies were estimated using PhyML[50] with LG substitution model and nearest neighbour interchanging during the tree search. We identified numerous novel viruses that clustered within the *Mitoviridae* in the phylogenetic tree, and these were excluded as they may have been infecting other organisms such as yeasts and were mostly uncommon.

**Viral abundance in RNAseq data**. Trimmed RNA reads from the 40 libraries were mapped to the same sequences as before (*Drosophila melanogaster*, *Wolbachia* and *Drosophila* related viruses, with all published sequences included) combined with the new viral contigs assembled from this population. Again, mapping was performed using STAR-2.6.0 with default settings (Supplementary Fig. 4; round.2 mapping). We counted the reads mapping to *D. melanogaster*, *Wolbachia* and each virus. Multiple mapped reads were counted once to a randomly selected mapped location.

The count of reads mapping to Grom virus (*D. obscura*) and Machany virus (*D. obscura*) read counts were positively correlated with that of their close relatives, respectively Motts Mill virus (*D. melanogaster*) and Kilifi virus (*D. melanogaster*)[22], suggesting miss-mapping (Supplementary Fig. 2D, E). Therefore, Grom virus and Machany virus read counts were reclassified into their respective relatives. Twyford virus was excluded from analyses as it is likely a virus of the fungal pathogen *Entomophthora muscae*[51]. Drosophila immigrans sigmavirus (DImmSV), which infects ~38% of *D. immigrans* flies[52], was excluded as there was evidence to suggest low levels of *D. immigrans* contamination in the RNAseq libraries, and the count of *D. immigrans* mitochondrial COI reads was positively correlated with the count of DimmSV reads (Supplementary Fig. 2C). Contamination could have arisen in the field, during collection or in the laboratory. In the most heavily contaminated library, the number of reads mapping to *D. immigrans* COI was <0.2% of the number of COI reads mapping to *D. melanogaster*.

We used our PCR data (see below) to identify pairs of contigs that were likely segments of the same viral genome. First, there was a strong correlation between the abundance of a new viral contig we identified and Vera virus ($r = 0.99$, $p < 10^{-10}$) (Supplementary Fig. 2A), suggesting these are two segments of the same Partitiviridae genome. The abundance of Galbut virus and Chaq virus were also strongly correlated ($r = 0.41$, $p < 10^{-10}$), but in this case many flies were infected with Galbut but not Chaq. This agrees with previous data suggesting Chaq virus is either a satellite virus of Galbut virus or an 'optional' segment of the Galbut virus genome[21]. We, therefore, refer to this sequence as Galbut (Chaq) virus.

**Virus prevalence**. Quantitative PCR (qPCR) was used to determine the presence and load of each virus in each sampled fly. Primers were designed in Primer-BLAST, which uses the Primer3 and BLAST, setting *Drosophila melanogaster* as the organism to check specificity[53,54]. For virus primer design, we used out RNAseq data to ensure there was no polymorphism in the first 5 bp in the 3′ end of each primer[55]. A degenerate base was used when a polymorphism was present elsewhere in the primer region with a minor allele frequency over 10%. No more than one degenerate site per primer was allowed. The efficiency with which each primer amplified viral RNA was estimated using a serial dilution of template cDNA. The complete list the primers, their efficiency, and the amplified product size can be found in Supplementary Table 1. To verify results for Vera and Motts Mill viruses we repeated the PCR tests of all the samples using an independent set of primers (Supplementary Fig. 2). Amplifications by qPCR were carried out with primer at a final concentration of 0.25 μM, using SensiFAST SYBR Hi-ROX master mix (Bioline) and 2 μL of a single-fly cDNA in a total volume of 10 μL. Reactions were performed in 96 well plates, including in each run six positive controls using cDNA library used in RNAseq as template and two template-free negative controls. The reactions were done using a StepOnePlus Real-Time PCR System in the following conditions: 95 °C for 2 min, 40 cycles at 95 °C for 5 s, 60 °C for 30 s. The product of the reaction was submitted to melting curve analysis to check the target-specific amplification, and samples where the melting curve was anomalous were discarded. To calculate relative viral load, we used the amplification of the host transcript *RpL32* (see above). Because primers for the viruses and the endogenous genes have approximate similar efficiencies, we calculated viral titer from the cycle thresholds ($Ct$) as $2^{\Delta Ct}$, where $\Delta Ct = Ct_{RpL32} - Ct_{virus}$.

**Statistics and reproducibility**. The effect of *Wolbachia* on the probability that flies were infected by viruses was estimated using a generalised linear mixed model implemented using the *R* package *MCMCglmm*[56], which uses Bayesian Markov chain Monte Carlo (MCMC) techniques. The binary response variable was whether or not a single fly tested positive for a given virus, which was treated as a binomial response with a logit link function. The model included a single fixed effect—whether or not a fly was infected with *Wolbachia*. The first random effect in the model was the identity of the individual fly being tested. The second random effect was the identity of the virus being tested for. For this random effect, separate variances were estimated for *Wolbachia*-infected and *Wolbachia*-free flies, and the covariance was set to zero (specified as 'idh(wolbachia):virus' in *MCMCglmm*). We used inverse Wishart priors ($V = 1$, $v = 0.002$). We estimated the prevalence of viruses in *Wolbachia*-infected and *Wolbachia*-free flies from the random effects of the model, and these estimates were transformed from the logit scale back into proportions. Credible intervals were obtained as the 95% highest posterior density of these random effects. To investigate if there was an effect of *Wolbachia* on flies being infected with a given virus, we calculated the proportion of samples from the MCMC chain where the viral prevalence in *Wolbachia*-infected samples is less than the prevalence in *Wolbachia*-free samples. The risk ratio was estimated by dividing the random-effects estimate of the prevalence in *Wolbachia*-infected flies by the estimate in *Wolbachia*-free flies for each sample from the MCMC chain, and then calculating the mean (posterior mean) and 95% highest posterior density (95% credible interval) of these numbers.

**Reporting summary**. Further information on research design is available in the Nature Research Reporting Summary linked to this article.

## Data availability

The RNAseq data have been submitted to the NCBI Sequence Read Archive under the BioProject number PRJNA728554. The assembled contigs of novel *D. melanogaster* associated viruses are available in GenBank (MZ852356 to MZ852369). The data underlying Figs. 1 and 2 are available in Supplementary Data 1 (Fig. 1), Supplementary Data 2 (virus prevalence), Supplementary Data 3 (risk ratios) and Supplementary Data 4 (viral load).

## Code availability

The code used for the bioinformatic analysis is available on the Github Repository at https://doi.org/10.5281/zenodo.552596857.

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

## Acknowledgements

We thank Darren Obbard for advice on viral taxonomy and nomenclature. Funding for this work was provided by a Newton Advanced Fellowship from the Royal Society, the São Paulo Research Foundation (FAPESP) (2013/25991-0 and 2015/08307-3), the National Council for Scientific and Technological Development (CNPq) (154568/2018-0 and 307447/2018-9) and the Natural Environment Research Council (NE/P00184X/1).

## Author contributions

R.C. and F.M.J. designed the study. R.C., A.C.P. and J.P.D. collected the data, S.D.D., F.M.J. and R.C. analysed the data. F.M.J. and R.C. wrote the manuscript with inputs from all other authors.

## Competing interests

The authors declare no competing interests.
