## [Transparent Peer Review File · Communications Biology]

Reviewers' comments:

Reviewer #1 (Remarks to the Author):

I don't have any substantive comments on this paper. It is well written, seems to be correctly analyzed, and provides data in an interesting area. The paper is a far stronger test of Wolbachia induced viral interference in a natural population of *Drosophila* than a previous effort which only had the potential to detect large effects such as seen in the blockage of dengue virus in mosquitoes. The current study makes it clear that any Wolbachia effects on virus load are likely to be quantitative so this requires the large sample sizes used here.

31 Is this really true? It has been known for a long time now that Wolbachia strains do not necessarily "manipulate" host reproduction and that these Wolbachia can have fitness benefits and spread in populations as evident from more recent reviews. This is later acknowledged in the paper such as for wAu in *Drosophila* (line 52 on).

66 1014 is impressive!

76-77. I found this sentence a little confusing. It might be worth having a Venn diagram of overlap between virus detections in different populations of *melanogaster* characterized so far. That way one could clearly see the differences in detections between studies and the overlap.
94 13 or 14 (see line 74).

General. It is worth thinking about sex effects. Males were tested in this paper, females in others. This is worth a comment.

Reviewer #2 (Remarks to the Author):

This manuscript demonstrates that Wolbachia is protective against viral infection in wild *Drosophila*. This was hypothesized on the basis of experimental lab studies, but previous field studies had been under-powered or poorly designed to detect the effect. Because the active release of artificially Wolbachia-infected mosquitoes is now an intervention strategy for the control of arboviruses such as dengue, this is a hugely important finding with a very broad audience.

This is a neat piece of work that tells a clear and simple story and needs relatively minimal revision – the majority of my suggestions below are for improvement (and to attract an even broader audience).

Major

1. Virus names – line 81 and elsewhere. The new virus names (ending in -like) are extremely unhelpful, especially when they end up with multiple "-like's ("Keenan toti-like virus-like 1"). This just creates an uninterpretable mess. Where do the authors imagine this is going to end? Who are they hoping will take responsibility for making sure we never have a "Keenan toti-like virus-like 1 - like 2 -like 3 -like7 -like" virus, if they are not willing to make the decision themselves? Just either give the viruses names or numbers "Drosophila totivirus 7" or pick a place name. Whatever the solution, make sure that none of the names and more than one "-like" in them.
2. The new segment of Vera virus is identified by co-occurrence with other segments in the pool-seq, and the presence of immigrants Nora sigmavirus is dismissed on the basis that it co-occurs with *D. immigrans* contamination. However, these analyses are not described in (any) detail and these data are never presented. (The immigrants sigmavirus result refers to a supporting Figure S3 that doesn't exist - S3 is something else). In addition, several of the viruses are thought to be infecting microbiota

rather than the flies themselves (e.g. Twyford virus is dismissed as a pathogen of a fungus, but this assertion is not tested when it could be). All of these issues require a more rigorous and transparent analysis of contaminating taxa. For example, but mapping reads to assembled COI sequences to demonstrate co-occurrence with fungal pathogens. This would be straightforward, and would not take long, but would strengthen the claim that many of the viruses are unlikely to be infecting the fly. Simply assigning host on the basis of phylogeny is extremely dubious: for example, all of the close relatives of Twyford virus are thought to infect insects (its an Iflavirus) rather than fungi.

3. L272-279. I am worried by the assertion that reads from *Drosophila immigrans* (and its sigma virus) are present "due to close proximity". This might be credible for DNA contamination, but seems unlikely for RNA. What mechanism do the authors propose? Are the reads in feces? It would be useful to know the actual read numbers. How many immigrans reads are there, versus how many immigrans sigmavirus reads (are both strands present?). Indeed, how were reads assigned to *D immigrans*? (this is not mentioned in the methods). Could the apparent presence of immigrans or immigrans sigmavirus reads could be mismapping (are they shorter reads? Unpaired? More errors?). L274 – I don't think reference 49 supports this statement on host range, as it presents no analysis of host-specificity. I'm not sure that any study has PCR-ed widely (i.e. large sample sizes) for drosophilid sigmaviruses in unexpected hosts. Perhaps <https://royalsocietypublishing.org/doi/10.1098/rspb.2016.2381> would be better, as it shows vertical transmission of this virus, but it doesn't do much to exclude the possibility of horizontal transmission.

4. Figure 1: The data presented in Panels 2 and 3 are not statistically compatible with each other. For example, Thika virus appears in all 40 pools of reads, but PCR found it to be present in only ~5% of flies. With 40 pools of 23 flies and a prevalence of 5%, the 99.9% CI is about 4-22 of 40 pools having at least one infected fly. The chance that all 40 contain a thika-infected fly is $p < 1e-6$. This means that the 100-read threshold for presence in panel 2 is too permissive, presumably due to barcode switching. (I am very surprised that barcode switching is not mentioned – could this explain the immigrans reads above? Did the libraries use single or dual indexed adaptors?). Given that it is possible to estimate prevalence from multiple pooled samples, I think it would be extremely useful to estimate prevalence for all of the viruses based on the pooled reads, and compare them to the prevalences inferred by PCR. The best solution for the barcode-switching might be to set flexible number thresholds based on a constant percentage of reads being misassigned. i.e. Count all the Thika reads across the whole dataset, and then set a % threshold that gives agreement with the PCR survey. In fact, now I think about it, the authors must know if any pools had reads for a virus but no flies in the pool PCR'd as positive for that virus. This would give a great estimate of read misassignment. Table S3 should include mention of pool membership.

5. Figure 2 – this is the central result of the paper, showing that *Wolbachia* is protective. However, the narrative emphasizes that it is protective against Chaq, Vera, and Motts Mill and that the viruses are different to each other, when in fact – although I recognize the viruses do differ – the overwhelming impression I get is that it is likely to be protective for most of the viruses, but they are not individually detectable. I think this analysis would be much better run as a mixed-effects model with virus as a random effect (a multi-membership model?). This would provide improved estimates of each virus (the random effects estimates could be interpretable) and would provide an overall estimate of the effect across viruses (and the variance among viruses).

6. It's a great shame that no attempt is made to speculate on the magnitude of the fitness benefits given the level of protection. A recent paper looks at the possible fitness consequences of Galbut virus <https://www.biorxiv.org/content/10.1101/2021.05.18.444759v1.full>. Just given the estimated protection against galbut/chaq or vera virus, how much fitness would this gain? Conversely, could this level of protection against galbut explain why its prevalence is <100%, despite ~100% transmission through both sexes?

Minor

1. L30 Does 2008 still count as recent?
2. L40-41 Is it really the case that no-one has looked at virus prevalence in the mosquitoes (comparing *Wolbachia* positive and negative?)

3. L49-50 There is a third, and more recent, study looking at the wolbachia/virus relationship in wild flies <https://www.biorxiv.org/content/10.1101/2021.05.09.443333v1>
4. L77-78 – That the melanogaster virome is nearing saturation sampling agrees with the comments of <https://doi.org/10.1098/rspb.2018.1165>
5. Supporting table 3 lacks essential information: What are the rows, and why are there 1111 of them?
6. Supporting table 1 should provide confidence intervals
7. L99 “The” Galbut virus – definite article is not necessary.
8. L103 and else where ‘sigma virus’ should probably be “*Drosophila melanogaster sigmavirus*” as the genus is now ‘sigmavirus’
9. Figure 2 figure legend is covered up, but I used copy+paste to read it.
10. L195-197 I worry that using a single host gene to normalize is a problem. Supposing having Wolbachia leads to a lower viral titer but this leads to healthier flies that have a higher level of rpl32 expression? I would worry a lot more if these were female, as infected flies might well lay fewer eggs, and rpl32 expression is very high in embryos. I doubt the effect in males would be big enough to be problematic, but would it also be possible to normalize to (e.g.) total RNA, or look at the correlations?
11. L88, L105 and many other places. ‘*Drosophila*’ shouldn’t be italicized when it’s part of the name of a virus. Or, if the virus species (as opposed to the virus) is being referred to, then the whole name should be italicized.

We appreciate all the comments and suggestion given by the two reviewers. Please find our point-by-point answers below in blue. All changes in the manuscript are highlighted in yellow.

Reviewer #1 (Remarks to the Author):

I don't have any substantive comments on this paper. It is well written, seems to be correctly analyzed, and provides data in an interesting area. The paper is a far stronger test of Wolbachia induced viral interference in a natural population of *Drosophila* than a previous effort which only had the potential to detect large effects such as seen in the blockage of dengue virus in mosquitoes. The current study makes it clear that any Wolbachia effects on virus load are likely to be quantitative so this requires the large sample sizes used here.

31 Is this really true? It has been known for a long time now that Wolbachia strains do not necessarily "manipulate" host reproduction and that these Wolbachia can have fitness benefits and spread in populations as evident from more recent reviews. This is later acknowledged in the paper such as for wAu in *Drosophila* (line 52 on).

This has been changed to:

"It infects approximately half of arthropod species, and many strains manipulate host reproduction, most commonly by inducing cytoplasmic incompatibility (CI)."

66 1014 is impressive!

76-77. I found this sentence a little confusing. It might be worth having a Venn diagram of overlap between virus detections in different populations of *melanogaster* characterized so far. That way one could clearly see the differences in detections between studies and the overlap.
94 13 or 14 (see line 74).

Previous studies have surveyed large numbers of populations with varying sample sizes, making it complicated to generate a figure comparing our data to previous studies. As this is a rather peripheral issue we have instead deleted the sentence.

General. It is worth thinking about sex effects. Males were tested in this paper, females in others. This is worth a comment.

We have added to the Discussion:

"As we cannot reliably morphologically identify female *D. melanogaster* to the species level, in this study we only investigated males. However, because Wolbachia is maternally transmitted, it is antiviral protection in females that will have the greatest effect on the symbiont's fitness and population dynamics. Therefore, an important question for the future is whether similar levels of antiviral protection are seen in female hosts."

Reviewer #2 (Remarks to the Author):

This manuscript demonstrates that Wolbachia is protective against viral infection in wild *Drosophila*. This was hypothesized on the basis of experimental lab studies, but previous field studies had been under-powered or poorly designed to detect the effect. Because the active release of artificially Wolbachia-infected mosquitoes is now an intervention strategy for the control of arboviruses such as dengue, this is a hugely important finding with a very broad audience.

This is a neat piece of work that tells a clear and simple story and needs relatively minimal revision – the majority of my suggestions below are for improvement (and to attract an even broader audience).

Major

1. Virus names – line 81 and elsewhere. The new virus names (ending in -like) are extremely unhelpful, especially when they end up with multiple '-like's ("Keenan toti-like virus-like 1"). This just creates an uninterpretable mess. Where do the authors imagine this is going to end? Who are they hoping will take responsibility for making sure we never have a "Keenan toti-like virus-like 1 -like 2 -like 3 -like7 -like" virus, if they are not willing to make the decision themselves? Just either give the viruses names or numbers "Drosophila totivirus 7" or pick a place name. Whatever the solution, make sure that none of the names and more than one "-like" in them.

We have renamed the viruses *Drosophila*-associated totivirus 1' etc. We opted to include the word 'associated' as we have not demonstrated these viruses infect *Drosophila* cells (although we would happily drop this if requested).

2. The new segment of Vera virus is identified by co-occurrence with other segments in the pool-seq, and the presence of immigrants Nora sigmavirus is dismissed on the basis that it co-occurs with *D. immigrans* contamination. However, these analyses are not described in (any) detail and these data are never presented. (The immigrants sigmavirus result refers to a supporting Figure S3 that doesn't exist - S3 is something else).

This supplementary figure was omitted by mistake in the submission process. These results are now shown in Figure S4.

In addition, several of the viruses are thought to be infecting microbiota rather than the flies themselves (e.g. Twyford virus is dismissed as a pathogen of a fungus, but this assertion is not tested when it could be). All of these issues require a more rigorous and transparent analysis of contaminating taxa. For example, but mapping reads to assembled COI sequences to demonstrate co-occurrence with fungal pathogens. This would be straightforward, and would not take long, but would strengthen the claim that many of the viruses are unlikely to be infecting the fly. Simply assigning host on the basis of phylogeny is extremely dubious: for example, all of the close relatives of Twyford virus are thought to infect insects (its an Iflavirus) rather than fungi.

We tried the suggested analyses but they proved to be not very informative. We took all the Trinity-assembled contigs with a blast hit ($E < 0.001$) against the Cytochrome c oxidase subunit 1 (COI) gene. These were collected and run through the Barcode of Life Data System (BOLD) (<http://www.boldsystems.org/>) to identify possible pathogen or fungal contamination. However, we only found a single short contig that could be a fungal pathogen (Zygomycota), so we did not pursue this analysis.

As several viruses are related to trypanosome viruses, we then searched for top blast hits that were from trypanosome genes. We took the longest putative trypanosome contig that had a top blast hit from *Leishmania*. The frequency of reads mapping to this contig did not correlate with the prevalence of any of our analyses. We chose not to report this result.

We have omitted Twyford virus as there is a published result that it infects a fungal pathogen of flies, not on the basis of our phylogenetic analysis.

3. L272-279. I am worried by the assertion that reads from *Drosophila immigrans* (and its sigma virus) are present "due to close proximity". This might be credible for DNA contamination, but seems unlikely for RNA. What mechanism do the authors propose? Are the reads in feces? It would be useful to know the actual read numbers. How many immigrants reads are there, versus how many immigrants sigmavirus reads (are both strands present?). Indeed, how were reads assigned to *D. immigrans*? (this is not mentioned in the methods). Could the apparent presence of immigrants or immigrants sigmavirus reads could be mismapping (are they shorter reads? Unpaired? More errors?). L274 – I don't think reference 49 supports this statement on host range, as it presents no analysis of host-specificity. I'm not sure that any study has PCR-ed widely (i.e. large sample sizes) for drosophilid sigmaviruses in unexpected hosts. Perhaps <https://royalsocietypublishing.org/doi/10.1098/rspb.2016.2381> would be better, as it shows vertical transmission of this virus, but it doesn't do much to exclude the possibility of horizontal transmission.

As discussed above we have now added the missing Figure S3 (now Figure S4) showing this data showing that DImmSV reads correlate with *D. immigrans* COI reads. This suggests we have contamination rather than a bioinformatic issue or new *Dmel* virus. Our speculation about host specificity or the source of the contamination is rather irrelevant (we did not mention lab contamination). Given that we have worked quite extensively with DImmSV this is a plausible source).

We felt excluding this virus was the conservative decision, but the evidence for contamination is also not very compelling so would be happy to add it back if the reviewer disagrees with that decision.

We have changed the text to avoid the speculation:

"*Drosophila immigrans* sigmavirus (DImmSV), which infects approximately 38% of *D. immigrans* flies, was excluded as the count of reads mapping to the *D. immigrans* mitochondrial genome was positively correlated with the count of reads mapping to DImmSV reads (Figure S4). This is consistent with contamination, which could have arisen in the field, during collection or in the laboratory. In the library with the most putative contamination, the number of reads mapping to *D. immigrans* COI was less than

0.2% of the number of COI reads mapping to *D. melanogaster*, so it is unlikely we misidentified any flies.”

4. Figure 1: The data presented in Panels 2 and 3 are not statistically compatible with each other. For example, Thika virus appears in all 40 pools of reads, but PCR found it to be present in only ~5% of flies. With 40 pools of 23 flies and a prevalence of 5%, the 99.9% CI is about 4-22 of 40 pools having at least one infected fly. The chance that all 40 contain a thika-infected fly is $p < 1e-6$. This means that the 100-read threshold for presence in panel 2 is too permissive, presumably due to barcode switching. (I am very surprised that barcode switching is not mentioned – could this explain the immigrans reads above? Did the libraries use single or dual indexed adaptors?). Given that it is possible to estimate prevalence from multiple pooled samples, I think it would be extremely useful to estimate prevalence for all of the viruses based on the pooled reads, and compare them to the prevalences inferred by PCR. The best solution for the barcode-switching might be to set flexible number thresholds based on a constant percentage of reads being misassigned. i.e. Count all the Thika reads across the whole dataset, and then set a % threshold that gives agreement with the PCR survey. In fact, now I think about it, the authors must know if any pools had reads for a virus but no flies in the pool PCR'd as positive for that virus. This would give a great estimate of read misassignment. Table S3 should include mention of pool membership.

We agree Panels 2 and 3 are incompatible, presumably due to 100 reads being too low. The spurious reads could arise from low-level contamination in the lab as well as barcode switching. This makes estimating a barcode switching rate from Thika virus and applying it to the viruses where we did not do single-fly PCR risky. In addition, some samples were drop from the single fly analysis after filtering out poor quality samples, further complicating the analysis.

Given that Panel 2 is both unreliable and unimportant to our conclusions, we have decided the conservative solution is to delete this Panel rather than statistically correct it.

Methods, added: “Adapters used were from KAPA Single-Indexed Adapter kits KK8701 and KK8702”

5. Figure 2 – this is the central result of the paper, showing that Wolbachia is protective. However, the narrative emphasizes that it is protective against Chaq, Vera, and Motts Mill and that the viruses are different to each other, when in fact – although I recognize the viruses do differ – the overwhelming impression I get is that it is likely to be protective for most of the viruses, but they are not individually detectable. I think this analysis would be much better run as a mixed-effects model with virus as a random effect (a multi-membership model?). This would provide improved estimates of each virus (the random effects estimates could be interpretable) and would provide an overall estimate of the effect across viruses (and the variance among viruses).

We have reanalysed the data with a mixed model, and the figures and text now reflect model estimates. When we added a Wolbachia x virus ‘interaction’ as a random effect, the overall main effect of *Wolbachia* on whether a fly tests positive for a given virus ceases to be significant, as might be expected if protection is virus-specific. We have not quoted a statistic on the variance among viruses as it was unclear to us what we could estimate that could easily be interpreted biologically. The specific changes are:

- Figure 2A and 2B have been replaced with estimates and error bars estimated from the random effects of the glm.
- Methods: The effect of Wolbachia on the probability that flies were infected by viruses was estimated using a generalised linear mixed model implemented using the R package MCMCglmm, which uses Bayesian Markov chain Monte Carlo (MCMC) techniques. The binary response variable was whether or not a single fly tested positive for a given virus, which was treated as a binomial response with a logit link function. The model included a single fixed effect—whether or not a fly was infected with Wolbachia. The first random effect in the model was the identity of the individual fly being tested. The second random effect was the identity of the virus being tested for. For this random effect, separate variances were estimated for Wolbachia-infected and Wolbachia-free flies, and the covariance was set to zero (specified as ‘idh(wolbachia):virus’ in MCMCglmm). We used inverse Wishart priors ($V=1$, $\nu=0.002$). We estimated the prevalence of viruses in Wolbachia-infected and Wolbachia-free flies from the random effects of the model, and these estimates were transformed from the logit scale back into proportions. Credible intervals were obtained as the 95% highest posterior density of these random effects. To investigate if there was an effect of Wolbachia on flies being infected with a given virus, we calculated the proportion of samples from the MCMC chain where the viral prevalence in Wolbachia-infected samples is less than the prevalence in Wolbachia-free samples. The risk ratio was estimated by dividing the random-effects estimate of the prevalence in Wolbachia-infected flies by the estimate in Wolbachia-free flies for each sample from the MCMC chain, and then calculating the mean (posterior mean) and 95% highest posterior density (95% credible interval) of these numbers.

- Results: We estimated the prevalence of each virus in *Wolbachia*-free and *Wolbachia*-infected flies, and found there are no cases where the symbiont completely blocks viral infection (Figure 2A). To quantify the level of protection we calculated the risk that a *Wolbachia*-free fly was infected with a virus relative to the risk of a fly carrying *Wolbachia* (Figure 2B). In 9 out of 10 cases the risk of infection was greatest in *Wolbachia*-free flies (Figure 2A and 2B), and for two viruses this effect was statistically supported (Figure 2A and 2B; $p_{mcmc} < 0.001$). These were a positive-sense RNA virus—Motts Mill virus—where the *Wolbachia*-free flies were 2.73 times more likely to be infected, and the dsRNA partitivirus Vera virus, where *Wolbachia*-free flies were 1.19 times more likely to be infected (Figure 2B). For both of these viruses we repeated the PCR tests of all the samples using an independent set of primers to verify these results (see Methods).
- Supplementary Data Files 2 and 3 have been revised with model estimates

6. It's a great shame that no attempt is made to speculate on the magnitude of the fitness benefits given the level of protection. A recent paper looks at the possible fitness consequences of Galbut virus <https://www.biorxiv.org/content/10.1101/2021.05.18.444759v1.full>. Just given the estimated protection against galbut/chaq or vera virus, how much fitness would this gain? Conversely, could this level of protection against galbut explain why its prevalence is <100%, despite ~100% transmission through both sexes?

We have added text speculating on fitness effects and *Wolbachia* dynamics. The reported transmission efficiency and virulence of Galbut virus are incompatible with the prevalence of the virus in the field. The magnitude of the effects we observe are insufficient to bridge this divide, so we have not discussed this.

- Discussion: The effect of *Wolbachia* on host fitness will depend not only on the reduction in viral prevalence and titre, but also on how harmful virus infection is to the fly. Of the three viruses affected by *Wolbachia*, only the phenotypic effects of Galbut virus infection have been reported. Under laboratory conditions this virus had only very modest effects on lifespan and fecundity³³. If we speculate that these results hold for other viruses affected, and given that *Wolbachia*-infected flies carrying ~0.37 fewer viruses, the magnitude of any fitness benefit might be so small as to have minimal impact on *Wolbachia* dynamics. However, harsh competitive conditions can increase the cost of infection, and these may be common in the field. For example, flies infected with the *Drosophila melanogaster sigma virus* appear healthy in the laboratory. However, in the field or under competitive laboratory conditions it is estimated to reduce fitness by 20-30%^{34,35}. If this was the case for the case for the viruses affected by *Wolbachia*, then the benefits of antiviral protection could be as high as 10%. This is comparable to the fitness benefit of *wAu* that allowed it to invade populations of *Drosophila simulans* in the absence of CI¹⁸.

Minor

1. L30 Does 2008 still count as recent?

I guess this depends how old you are ... deleted.

2. L40-41 Is it really the case that no-one has looked at virus prevalence in the mosquitoes (comparing *Wolbachia* positive and negative?)

We are unaware of such a study being done.

3. L49-50 There is a third, and more recent, study looking at the *wolbachia*/virus relationship in wild flies <https://www.biorxiv.org/content/10.1101/2021.05.09.443333v1>

Thanks for spotting this omission. This section now reads:

Our results contrast with three previous that failed to find any effect of *Wolbachia* on the natural viral community of *Drosophila*. The first of these was a study designed to characterize the diversity of viruses infecting *D. melanogaster* and *D. simulans*, and the authors suggest their sampling design means they have low power to detect the effects of *Wolbachia* 16. The second study investigated *D. melanogaster*, but used considerably smaller sample sizes than us and reared the flies for one or more generations in the laboratory at 19°C before testing them 15. It was later discovered that the antiviral effect of *wMel* is greatly reduced at this temperature 33. Finally, another study investigated *D. simulans* but used comparatively small sample sizes that are unlikely to detect effects of the size we observed.

4. L77-78 – That the *melanogaster* virome is nearing saturation sampling agrees with the comments of <https://doi.org/10.1098/rspb.2018.1165>

This section is now deleted in response to Reviewer 1.

5. Supporting table 3 lacks essential information: What are the rows, and why are there 1111 of them?
Sorry, this file was neither clear nor correct. (1) Some control samples had been included. These have been removed leaving 1014 samples. (2) An extra sample ID column has been added and a brief explanation of what the numbers are has been added to the start of the file.
6. Supporting table 1 should provide confidence intervals
Added and table headers made clearer
7. L99 “The” Galbut virus – definite article is not necessary.
Corrected
8. L103 and else where ‘sigma virus’ should probably be “Drosophila melanogaster sigmavirus” as the genus is now ‘sigmavirus’
Corrected
9. Figure 2 figure legend is covered up, but I used copy+paste to read it.
Corrected
10. L195-197 I worry that using a single host gene to normalize is a problem. Supposing having Wolbachia leads to a lower viral titer but this leads to healthier flies that have a higher level of rpl32 expression? I would worry a lot more if these were female, as infected flies might well lay fewer eggs, and rpl32 expression is very high in embryos. I doubt the effect in males would be big enough to be problematic, but would it also be possible to normalize to (e.g.) total RNA, or look at the correlations?
We have checked the result holds without normalising the data and added:
“Furthermore, this result still holds if the viral load was not normalised to rpl32 mRNA levels, indicating that it is not an artefact of Wolbachia affecting expression of the reference gene we used ($F=14.47$, $d.f.=1,632$, $p=0.0002$).”
11. L88, L105 and many other places. ‘Drosophila’ shouldn’t be italicized when it’s part of the name of a virus. Or, if the virus species (as opposed to the virus) is being referred to, then the whole name should be italicized.
Corrected

REVIEWERS' COMMENTS:

Reviewer #2 (Remarks to the Author):

See my previous comments for context. The authors have addressed the majority of my concerns – just two points arise from the changes:

1) In the description of the statistical methods, virus is described as a random effect. How does the model deal with flies that have more than one viral infection? I think the analysis code for this model needs to be made available, perhaps in supporting material.

2) Looking at figure S4C.

First, there is a presentation issue with this axes to this figure that needs correcting.

Second, it appears (if the two axes are on the same scale?) that the sample most contaminated with Dimm sigma virus has nearly 5000 sigma reads per million mapped reads, but the sample most contaminated with *D. immigrans* has only 2.5 *immigrans* reads per million. This seems to imply that contaminating sigma reads are roughly 2000 times more numerous than contaminating COI reads. Indeed, if Dimm COI in that sample is 0.2% of *Dmel* COI (as quoted) it seems that there is 4 times more contaminating Dimm sigmavirus than there is *Dmel* COI. Is this really compatible with contamination? Perhaps it is just a length normalisation issue?

1) In the description of the statistical methods, virus is described as a random effect. How does the model deal with flies that have more than one viral infection? I think the analysis code for this model needs to be made available, perhaps in supporting material.

-We have added the code to <https://doi.org/10.5281/zenodo.5525968>

2) Looking at figure S4C.

First, there is a presentation issue with this axes to this figure that needs correcting.

Second, it appears (if the two axes are on the same scale?) that the sample most contaminated with Dimm sigma virus has nearly 5000 sigma reads per million mapped reads, but the sample most contaminated with D immigrans has only 2.5 immigrans reads per million. This seems to imply that contaminating sigma reads are roughly 2000 times more numerous than contaminating COI reads. Indeed, if Dimm COI in that sample is 0.2% of Dmel COI (as quoted) it seems that there is 4 times more contaminating Dimm sigmavirus than there is Dmel COI. Is this really compatible with contamination? Perhaps it is just a length normalisation issue?

- We fixed the issue with the axes.

Yes, the virus is almost 10 times the length of the COI gene, so the copy number of the virus+virus transcript is ~200 times that of COI transcripts. In the absence of data of this number in D. immigrans we cannot comment on how plausible this is. Furthermore, packaged viral RNA may be more stable than mRNA, which will also affect this ratio. In our view excluding this virus is the conservative course of action.